# Foliar Application of dsRNA Targeting Endogenous Potato (*Solanum tuberosum*) Isoamylase Genes ISA1, ISA2, and ISA3 Confers Transgenic Phenotype

**DOI:** 10.3390/ijms24010190

**Published:** 2022-12-22

**Authors:** Ido Simon, Zohar Persky, Aviram Avital, Hila Harat, Avi Schroeder, Oded Shoseyov

**Affiliations:** 1Robert H. Smith Faculty of Agriculture Food and Environment, Hebrew University, Rehovot 76100, Israel; 2Department of Chemical Engineering, Technion—Israel Institute of Technology, Haifa 32000, Israel

**Keywords:** potato, starch, sucrose, sugar, RNAi, gene silencing, SIGS, siRNA, lmPEI, dsRNA

## Abstract

Isoamylase (ISA) is a debranching enzyme found in many plants, which hydrolyzes (1-6)-α-D glucosidic linkages in starch, amylopectin, and β-dextrins, and is thought to be responsible for starch granule formation (ISA1 and ISA2) and degradation (ISA3). Lipid-modified PEI (lmPEI) was synthesized as a carrier for long double-stranded RNA (dsRNA, 250-bp), which targets the three isoamylase isoforms. The particles were applied to the plant via the foliar spray and were differentially effective in suppressing the expressions of ISA1 and ISA2 in the potato leaves, and ISA3 in the tubers. Plant growth was not significantly impaired, and starch levels in the tubers were not affected as well. Interestingly, the treated plants had significantly smaller starch granule sizes as well as increased sucrose content, which led to an early sprouting phenotype. We confirm the proposal of previous research that an increased number of small starch granules could be responsible for an accelerated turnover of glucan chains and, thus, the rapid synthesis of sucrose, and we propose a new relationship between ISA3 and the starch granule size. The implications of this study are in achieving a transgenic phenotype for endogenous plant genes using a systemic, novel delivery system, and foliar applications of dsRNA for agriculture.

## 1. Introduction

The potato is an extremely important staple crop for humans, is grown and consumed in vast quantities, and is the only staple tuber crop grown outside of the tropics [1]. Potatoes account for approximately 45% of total worldwide tuber production [1], at approximately 320 million tons, and are grown for a multitude of downstream purposes, such as starch, alcohol production, feed, food, and biotechnology applications [2,3]. Lastly, potatoes serve as a valuable model system for studying tuber development and mechanisms underlying starch synthesis [4].

Potatoes are carbohydrate-rich tubers, which contain approximately 80% water and 20% dry matter, of which, 60–80% of the dry matter consists of starch [2]. Although starch biosynthesis and regulatory genes have been extremely well-studied, the actual regulation of starch synthesis has been proven to be much more complex than what may be occurring at the transcriptional level. In the initial stages of tuber formation, starch synthesis genes experience a strong increase in expression, due to a metabolic flux from apoplastic toward symplastic sucrose loading, which is maintained until the final tuber size is reached [4]. In short, sucrose passes from the phloem to the cytosol, undergoing a series of reactions where it is enzymatically broken down into Glu-6-P—at which point it is imported to the amyloplast and converted into ADP glucose, the building block of starch. From there, soluble starch synthase and branching enzymes synthesize amylopectin and amylose. In addition to the starch-modifying enzymes, there is also differential regulation between the isoforms of isoamylase, a starch-debranching enzyme [4,5,6].

Starch granule initiation is believed to be due to isoamylase activity [5]. There are three distinct, evolutionarily conserved isoforms of isoamylase (ISA) in several plants, including potatoes, wheat, and *Arabidopsis* [5]. Current data suggest that ISA1 and ISA2 act together in a multimeric enzyme to initiate starch granule synthesis and crystallization, while ISA3 is not associated with the enzyme complex and is thought to degrade starch according to current literature [5,6,7]. Starch in plants is composed of two types of α-1,4-linked glucan polymers: unbranched amylose and branched amylopectin, which contain the α-1,6 branch points introduced by starch-branching enzymes [6]. The activity of starch de-branching enzymes, including the isoamylase, which hydrolyze the α-1,6 linkages, is also thought to be critical in the synthesis of amylopectin through the allowance of further crystallization sites with the hydrolysis of the branched linkages [5,6]. These relationships are further complicated by the differential expression of the genes in combination with multimeric complexes and different catalytic specificities of each isoform [4,6]. The relationship between ISA1, ISA2, and ISA3 still is not fully understood. For instance, an *Arabidopsis ISA3* loss of function mutant exhibited a starch-excess phenotype, while the simultaneous suppression of ISA1, ISA2, and ISA3 by transgenic hairpin RNAi resulted in a starch-deficient phenotype, despite the suppression of ISA3 which, according to the current literature, is responsible for degrading starch [4,5,6,7].

Ever since the development of the first genetically-modified (GM) crop in the 1990s, commercial GM traits have been produced in 15 crop plants, and in 2015, over 18 million farmers from 28 countries planted almost 440 million acres of GM crops [8]. Most commercial GM crops today are modified for insect resistance and herbicide tolerance traits, mostly due to the long regulatory process involved, high costs, and a primary challenge in commercialization of agronomic traits consistently delivering performance improvements across diverse genetic backgrounds and diverse environmental conditions [8].

In improving the plant performance, not the insect or pathogen resistance, in order to deliver traits with sufficient impacts to be economically viable, development should take into account germplasm and environmental niches and, thus, requires compensatory decreases in other product development costs [8]. The need for improved plant performances through GM is only relevant when non-modified plants lack certain characteristics or responses to environmental conditions and are not optimized to maximize economic yield under those conditions. This greatly reduces the number of profitable GM targets the market is willing to pay for [8]. Bringing a transgenic crop from development to commercialization can cost as much as USD 140 million of the standard transgene delivery methods and can lead to off-target effects, [9] and public opposition to GM crops is widespread [10]; all of these factors are where the strategy of spray-induced gene silencing (SIGS) can be advantageous.

SIGS is the topical application of double-stranded RNA molecules (dsRNA), which has the advantage of cheaper manufacturing of source RNA and the ability to rapidly customize and synthesize various gene targets (and not be considered genetically modified) [10]. SIGS not only improves plant traits and agronomic performance, it can also target crop pests and diseases, which account for most of the traits used in GM crops today [10]. In addition to targeting pests and diseases, SIGS can also be used on endogenous traits of interest with commercial viability, such as the targeting of a glutathione S-transferase gene in grapevines, which increases drought resilience [11]. SIGS functions on the basis of RNA interference (RNAi), which is a conserved eukaryotic process in which transcript accumulation is modulated or reduced in a sequence-specific manner due to small, non-coding RNAs (siRNA) between 20 and 40 nucleotides in length [10]. A core set of proteins, such as Dicer and Argonaute, are involved in processing the dsRNA into siRNA molecules, which move to neighboring cells across plasmodesmata or apoplectically through intercellular spaces [10]. These siRNAs are also capable of long-distance and systemic transport in plants [9,10].

However, there are many issues facing the commercialization of SIGS technology. One of which is that application of naked dsRNA is prone to RNA degradation and poor penetration, as well as shortened silencing duration and non-specificity [9,10]. Some of these issues can be alleviated via the delivery system as the one used in this study, which uses rapidly synthesized and scalable dsRNA-lmPEI particles exhibiting temperature stability, protection from ribonuclease activity, and specific delivery characteristics [9]. There are still many unknowns and variables in regard to SIGS. Nanoparticle delivery and efficacy [12], optimal physiological plant conditions [13], and efficient gene targeting [10] are all areas that need to be further explored. For instance, it was recently found that plants cannot uptake nanoparticles larger than 20 nm; however, cellular internalization of the nanoparticles themselves is not required, only the intercalation of molecules between cells is required for efficient cargo delivery and release [12]. The time of day that the dsRNA is applied and the soil moisture of each pot can also significantly affect gene silencing efficacy [13]. Lastly, the pressure at which the dsRNA spray is applied can also have a large effect on the efficacy of the treatment [14].

Here, we report on the simultaneous targeting of ISA1, ISA2, and ISA3 using a single spray with a lmPEI nanoparticle dsRNA carrier and SIGS approach. A transgenic phenotype similar to previous research with a transgenic chimeric construct [7] was achieved, albeit with reduced gene silencing efficiency and no changes to the starch concentration. Despite no changes to the starch concentration, an increased sucrose concentration was observed as well as an early sprouting phenotype. Detailed analysis of the gene expression revealed differential silencing using the SIGS approach, as well as a potential influence of ISA3 on starch granule sizing.

## 2. Results

### 2.1. SIGS Differentially Suppresses ISA1, ISA2, and ISA3 Expression in Leaves and Tubers

Leaves treated with the gene-silencing spray targeting ISA1, ISA2, and ISA3 had variable reductions in gene expressions, and in the case of ISA3, no effect at all in the leaves. As seen in Figure 1, both ISA1 and ISA2 were significantly suppressed to about 50% of gene expression 2 weeks after the first spray. After 4 weeks and 2 sprays, the gene activities of both returned to normal, only to be significantly reduced again to approximately 60% after 6 weeks and 3 sprays. After 4 sprays and 8 weeks, ISA2 began to return to expression levels similar to the control group, while ISA1 continued to be suppressed throughout the entire growth cycle. The gene expression of ISA3 in the leaves was not significantly affected throughout the entirety of the growth cycle.

Surprisingly, gene suppression in the tubers did not match the suppression activity of the spray in the leaves. As seen in Figure 2, the only significant reduction in gene activity was for ISA3 (25%, *p* ≤ 0.05), which was not suppressed at all in the leaves as shown in Figure 1. However, although not significant, there was a trend in the reduced activity of ISA1 (13%, *p* = 0.19) in the tubers as well.

### 2.2. Suppression of ISA Genes Does Not Significantly Alter the Potato Yield

The suppression of ISA1 and ISA2 in the leaves and ISA3 in the tubers did not significantly affect the yield, although there was a trend in the treated plants toward fewer potatoes and a median potato weight almost twice that of the control (*p* = 0.06 and 0.1, respectively) as seen in Figure A1.

Visually, both treated and control plants were identical; no phenotypic changes or differences in growth habits were observed.

### 2.3. ISA3-Suppressed Potato Tubers Do Not Exhibit Changes in Starch and Protein Concentrations, but Show Significantly Higher Sucrose Concentrations

Treated and control potatoes were processed in parallel and total starches as percentages of fresh weight were measured spectrometrically using iodine. There were no significant differences between total, soluble, and insoluble starch concentrations in the tubers (data are shown only for the total starch in Figure 3).

As shown in Figure A2, there was also no change to the total protein as a percent of fresh weight in the treated plants. The SDS-PAGE analysis of the protein samples also showed no difference in the strength of the 44 kDa bands for patatin, which is the main protein of interest in potatoes [3].

Interestingly, despite no reduction in starch content in the treated plants, there was a significant increase in the total sugars present in the tubers as determined by the anthrone method and reported in sucrose equivalents. As mentioned, the lyophilized potatoes were extracted with ethanol, which does not dissolve the starches and only dissolves the sugars present [15]. As seen in Figure 4, most of the control samples were at or below 13 µmol per gram FW of tissue. The treated samples had on average 15% more sucrose, although there was greater variability in the treatment group with the highest concentrations being in the range of 18–20 µmol of sucrose per gram FW of tissue.

**Figure 3 ijms-24-00190-f003:**
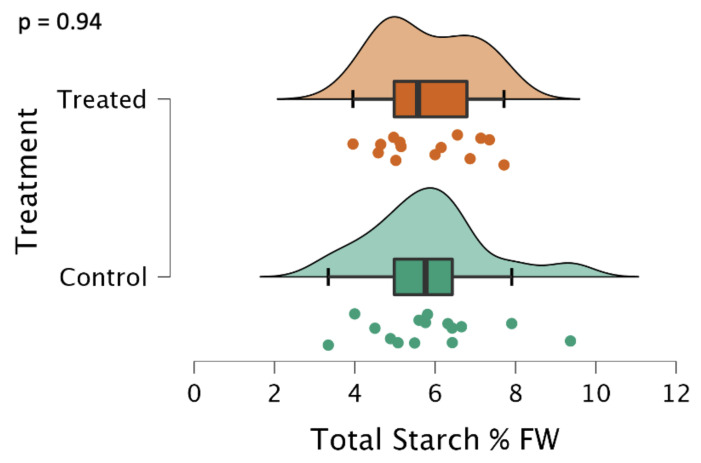
Total starch as a percentage of the tuber’s fresh weight, represented as a raincloud plot [16]. The graph is comprised of a boxplot with the lines representing the median, lower, and upper-end measurements, as well as lower and upper quartiles. Each dot represents the recorded value of the biological replicate, with the cloud representing the overall distribution of the data. Note that there is no significant difference, *p* = 0.94 (n = 15, in triplicate, each for treatment and control). The total starch was determined spectrometrically using a calibration curve with potato starch and the USDA rapid starch method [17].

### 2.4. ISA3-Suppressed Potato Tubers Exhibit Significantly Reduced Starch Granule Sizes

Most surprisingly, the treated tuber group, which did not show a significant reduction in ISA1 expression, but did show a significant reduction in ISA3 expression (Figure 2) still exhibited reduced starch granule sizes as seen in Figure 5. There was a 25% reduction in the median granule size (*p* = 0.05), and a large shift in the distribution of particle sizes toward less than 400 pixels^2^ as determined by the ImageJ analysis. On average, 1000 particles per biological replicate were examined, for a total of about 29,000 particles analyzed for the entire experiment.

Representative micrographs (Figure 6 from the control and treated groups are included below. Micrographs were photographed at 4× optical zoom. Note the increased counts of smaller starch granules observed in the treated group, as well as a reduced count of extremely large particles.

The cumulative starch granule distribution shifted slightly to the left as well, with 70% of particles from treated tubers having a size of 750 pixels or less, and 70% of particles from control tubers having a size of 1000 pixels or less, as seen in Figure 7.

When looking at the non-cumulative granule distribution in Figure A3, we can see that the distribution shapes of the treated and control tubers are similar, as are the largest occurrences of particle species, around 500 pixels^2^ for both control and treated tubers. However, treated tubers have an increased amount of granules between 100 and 150 pixels^2^ in size and a reduced amount of granules above 750 pixels^2^ in size.

### 2.5. Treated Potato Tubers Exhibit Early Sprouting behavior

The treated potatoes exhibited early sprouting behavior, as seen in Figure 8. Interestingly, some control potatoes began sprouting before the treated potatoes; however, the treated potatoes started sprouting at a more rapid pace after 50% of the potatoes had sprouted. The largest difference between the two groups was one week for the treated potatoes at 80% sprouting.

The number of sprouts was significantly higher (*p* ≤ 0.05) in the treated potatoes; the median number of sprouts treated was approximately four, while the control potatoes had a median of two, as seen in Figure 9. Interestingly, the variance in this measurement is extremely high, ranging from two sprouts per tuber to eight sprouts per tuber, whereas in the control, there were no more than four sprouts per tuber in all of the samples.

### 2.6. Data Robustness

In order to test the data robustness, linear regressions were performed for the recorded variables in this experiment in order to explore their relationships with one another, as seen in the Pearson correlation heatmap in Figure A4. Significant relationships were determined between the number of potatoes and median potato weight (−0.713, *p* ≤ 0.05), water content, and the average weight of the whole plant (0.411, *p* ≤ 0.05), the total starch content and number of potatoes (0.453, *p* ≤ 0.05), total protein, and total sugar (0.404, *p* ≤ 0.05), and total sugar and number of sprouts (0.382, *p* ≤ 0.05).

## 3. Discussion

This is the first occurrence, to our knowledge, where a gene in a tuberous crop was successfully suppressed through the exogenous application of dsRNA to the leaves, although the gene silencing observed was not as efficient as transgenic methods [7]. Furthermore, a transgenic phenotype was achieved in the tubers, despite the lack of constitutive expression of dsRNA, with only bi-weekly applications of dsRNA.

We observed a phenotype similar to that observed when ISA1, ISA2, and ISA3 were silenced with transgenic hairpin RNAi [7], although we did not suppress ISA1 and ISA2 in the tubers, which is likely why the treatment was also ineffective at reducing starch content (Figure 3). Despite this, there was still a significant reduction in the median granule size (Figure 5), an increase in sugar levels (Figure 4), a decrease in the sprouting time (Figure 8), and an increase in the number of sprouts (Figure 9). No differences in water content were observed; however, there were trends in the treated phenotype (Figure A1) being a smaller amount of potatoes per plant (*p* = 0.06) and larger median potato weight (*p* = 0.10).

In the current literature, there are many examples of SIGS for use against viruses, fungi, and insects [10]. However, there are only a few examples of SIGS being utilized for endogenous gene silencing [11,19] due to various challenges. These include RNA instability [9,10], barriers to particle uptake into cells [12], and application methods required for efficient silencing [13,14]. In this experiment, we attempted to silence endogenous potato genes using a dsRNA-lmPEI nanoparticle carrier [9], which was previously used to confer viral disease resistance in grapevines. The lmPEI carrier was shown to have systemic distribution, where lower concentrations of the particles were found in the roots. As the dsRNA was processed into siRNAs, we believe they migrated toward areas of higher expression for the specific genes—ISA1 and ISA2 in the leaves, and ISA3 in the tuber [6], which resulted in the differential gene silencing observed.

The continued suppression of ISA1 and subsequent increase of ISA2 expression observed in the leaves (Figure 1) also reaffirms the complex relationship between the isoamylase enzymes [5,7]. It is thought that ISA2 is responsible for regulating the activity of ISA1 through a multimeric complex [5,6,7], which matches the observed expression data, especially when taking into account that this experiment did not use a transgenic promoter for continual suppression of the genes. Multiple administrations of the spray showed a continued decrease in gene expression. ISA1 and ISA2 were suppressed in leaves by 50% two weeks after the first treatment, but returned to normalcy four weeks after the first treatment, which coincides with tuber initiation in potatoes [4]. ISA2 was briefly suppressed after the third treatment until week six and returned to normal gene expression as ISA1 continued to be suppressed from week six to week ten.

The sequence choice likely also played a role in the efficacy of gene suppression observed [19]. One study examined the locations of various genes within the potato and found that the three isoamylase genes are spread across different chromosomes and that there are multiple isoforms of ISA1 [20]. This leaves open the possibility that siRNAs are ineffective at reaching certain parts of the plant, and also that the chosen dsRNA sequence targeting ISA1 is ineffective in silencing multiple isoforms, which could be one explanation for the suppression activity seen in leaves and the lack of suppression of ISA1 in the tubers. One such example of this is in the SIGS of *Arabidopsis thaliana* genes, where it was found that one targeted gene was successfully silenced via siRNAs derived from exogenous dsRNA, while another targeted gene’s siRNAs were degraded and the gene was not effectively silenced [19].

Literature studies today attribute decreased starch levels, granule sizes, and increased sucrose to the silencing of ISA1 and ISA2 [5,6,7]. The gene expression data observed in this trial partially support this conclusion. ISA1 and ISA2 were suppressed in the leaves; however, this was not reflected in the tuber. Given this, we are proposing that ISA3 also has a relationship to starch granule sizing in the *Solanum tuberosum* tuber, which to the best of our knowledge, is not present in the current literature. Additionally, there has not been a mutant analysis of ISA3 in potatoes, only a complete loss of the function mutant, *isa3* in *Arabidopsis thaliana*, which observed a starch-excess phenotype in its leaves [5,6]. Given that the role of ISA3 is thought to degrade and mobilize starch, one would expect to see a starch-excess phenotype in our experiment. One possible explanation for the lack of this phenotype is that smaller starch granules are more easily turned over into sucrose, which was suggested in a previous experiment that silenced the three isoamylase genes via hairpin RNAi [7]. We propose that suppressing ISA3 activity in the tuber is impactful enough to reduce granule sizing without affecting its ability to degrade or mobilize starch, thus not resulting in the starch-excess phenotype previously observed with its loss of function. It is important to also consider an alternative explanation, where the suppressed ISA1 and ISA2 genes in the leaf had roles in the tuber phenotype or allowed ISA3 to affect granule sizing. Given the complex relationships between the isoamylase genes [5,6,7], these relationships should be further explored in future research through the use of targeted transgenic methods in *Solanum tuberosum*.

In this work, we tested the hypothesis that a lower content of starch, achieved through the suppression of the three isoamylase genes, would lead to a metabolic flux toward sucrose sinks, more aromatic amino acid production and a higher protein concentration in the potato tuber. The results show a good correlation between the total sugar content and total protein (Figure A4). Interestingly, we did not observe an increase in total protein (Figure A2) despite observing an increase in total sugar (Figure 4). This could potentially be explained by the changing sugar needs and translocation of sugars throughout the life cycle of the potato plant. There are many sinks that the sugar could flow to [4,20], and given the variability of the potato plant, the additional protein sink may not adequately increase to produce a statistically significant difference, despite the fact that significantly more sugar was observed in the treated tubers. However, the increased sugar was reflected in the doubling of the number of sprouts observed in the treated tubers (Figure 9), which once again confirms the phenotype observed in this study and prior literature [7].

## 4. Materials and Methods

A full experimental outline summarizing the number of biological and technical replicates is included in Figure A5.

### 4.1. Statistical Analysis

All statistical analyses were performed using JASP [21] with one-way analysis of variance (ANOVA) tests and a significance level of *p* ≤ 0.05. Significance for the cumulative granule size distribution was determined by Kolmogorov–Smirnov tests for the equality of distributions and a significance level of *p* ≤ 0.05. For testing the differences in the means, a regression analysis was performed for univariate and robustness, as well as a multivariate regression, which controlled for the number of potatoes, median potato weight, and water content on a per-plant basis. We controlled for these variables because they are variables likely to influence the measured outcomes and at the same time could be affected by the treatment. For the same reasons, similarly sized and shaped potatoes were chosen for the analysis. Lastly, a correlation heatmap was created between the number of potatoes, median potato weight, average weight of the whole plant, water content, total starch, total protein, total sugar, median granule size, and the number of sprouts on a per-plant basis.

### 4.2. Plant Material and Sampling

*Solanum tuberosum* L. cv. ’Desiree’ were vegetatively propagated from sprouting tubers in a mixture of 50% universal potting soil (Tuff, Merom Golan, Israel) and 50% perlite #4 (Agrekal, Moshav Habonim, Israel) with day temperatures of 23 °C and nighttime temperatures of 20 °C in a greenhouse with natural lighting conditions from the months of February to May. Plants were individually grown in 4 L pots with 20–20–20 fertigation through drip irrigation three times per day. Irrigation was stopped at week 12 of growth until the leaves were desiccated. Tubers were harvested at week 15, and stored in darkness at room temperature. Leaves were sampled bi-weekly for RNA prior to each treatment application and flash-frozen in liquid nitrogen.

Every two weeks, for a total of six sprays over a 15-week growth period, the treated plants were sprayed. In order to ensure consistency between the sprays, the dsRNA-lmPEI solutions were vigorously vortexed and the three individual solutions for each gene were mixed into the handheld sprayer (Solo, Newport News, VA, USA). The plants were sprayed at the same time of day until seepage was reached on each individual plant. Additionally, RNA sampling was performed prior to the bi-weekly spray in order to judge the effect of the prior spray and also to ensure that there was not a localized high concentration of dsRNA on the flash-frozen leaves, which could interfere with the RT-qPCR analysis or prevent the slow-release of dsRNA into the plant after the spray.

At harvest, tubers were characterized and weighed individually, then similarly-sized and shaped tubers were split into groups to monitor sprouting behavior and metabolite analysis. Tubers of sizes 20–60 g and similar shapes were taken for metabolite analysis, while larger tubers were separated for sprouting behaviors.

For metabolite analysis, approximately 80 g of tubers per biological replicate were weighed, washed, and dried with paper towels. The tubers were then cut into slices, blended using an immersion blender (Newfan, Rishon Lezion, Israel), and split for the analysis of the starch granule size, protein, starch content, flash frozen for RNA extraction, and lyophilized for water content and sugar determination. All materials for RNA were stored at −80 °C until extraction.

### 4.3. dsRNA Design and Preparation of dsRNA-lmPEI Particles

In order to target the three forms of the isoamylase gene ISA1, ISA2, and ISA3, 250 bp-long sequences were chosen (shown below) in order to trigger the dicer-like protein at different locations along the protein and generate multiple siRNAs as previously described [9]. All three sequences were examined for their specificity using NCBI’s BLAST tool [22].

ISA1:(NCBI accession number NM_001288008).TTT TTT GAG ATT TTG CGG CCT CAT GAC CAA ATT CCG CCA TGA ATG TGA ATC ACT GGG ATT AGA TGG TTT CCC TAC AGC AGA AAG GCT GCA ATG GCA TGG TCA CAC TCC TAG AAC TCC AGA TTG GTC TGA AAC AAG TCG ATT CGT TGC ATT CAC ACT GGT CGA CAA AGT GAA GGG AGA ACT ATA TAT TGC CTT TAA CGC CAG CCA TTT GCC TGT AAC GAT TAC ACT TCC AGA TAG GCC TGG TTA TAG ATGGISA2: (NCBI accession number XM_006355814).CCC CTG CAT ATG ATG CTC GAA AAT CTT TGG GTT GGA ATA CTT TAA AAA CTG GTT TTG GGA CTC AGA TTG CCC AGT TTA TTT CAT TCT TGA GTA ATT TAA GAA TGA GAA GAA GTG ATC TTC TTC AAA AGA GAA CCT TCT TGA AGG AAG AAA ACA TCC AGT GGC ATG GGA GTG ACC AAT CTC CTC CGA AAT GGG ATG GCC CGT CTA GCA AAT TCT TGG CTA TGA CTT TGA AGG CCG ATG CTG AAG TCA GCCAISA3: (NCBI accession number NM_001288291).ATG ATG CAA ACG GTG AAG GTG GCA ATG ATG GAT GCA ATG ACA ACT TCA GTT GGA ATT GTG GAA TTG AAG GTG AAA CTT CAG ATG CAA ATA TTA ACG CAC TGC GTT CAC GGC AAA TGA AAA ATT TTC ATT TGG CAC TGA TGG TTT CTC AGG GAA CAC CAA TGA TGC TTA TGG GGG ATG AGT ATG GGC ATA CCC GCT ATG GAA ATA ATA ACA GTT ATG GAC ATG ATA CCG CCA TCA ACA ATT TCC AGT GGGG

The chosen dsRNA sequences were synthesized, ordered (AgroRNA, Seoul, Republic of Korea), and complexed with lmPEI, as previously described [9]. In short, 14-carbon lipids were conjugated to branch PEI to form lmPEI, which was complexed with the 250 bp dsRNA under acidic conditions. Nanoparticle characteristics were examined before each experiment and did not vary from the characteristics previously described [9]. Although the absorption of dsRNA into the plant was not determined in this specific experiment, prior experiments utilizing Gd-lmPEI in grapevines showed systemic absorption in the plant [9].

The dsRNA-lmPEI particle stability was also previously tested [9] using gel agarose, was found to be stable against RNase, and released from the lmPEI with heparin. This stability previously conferred 1–2 weeks of reduced grapevine leafroll virus titer after application in grapevines [9].

### 4.4. RNA Extraction and Quantitative Real-Time PCR (RT-qPCR)

RT-qPCR was performed with 15 biological replicates for each control and treatment group, and at least 3 technical replicates unless otherwise stated. Expression levels for the three isoamylase genes and reference gene ubiquitin (*ubi3*, L22576.1) were determined by RT-qPCR with the primers described below [4,7]. The amplicon sizes and primer efficiencies are included in Table A1.

ISA1: 5′ to 3′Forward: GGCAAATGGAGAGGACAACAReverse: ATGGGAACACCTTGGGAAACISA2: 5′ to 3′Forward: TTATCCTTCCGCCACCTCReverse: CTTCAACTGGAGTTCCCTTCTISA3: 5′ to 3′Forward: GACGCTTGCCCTTCATTCReverse: CTCCTGTGCGGTTCTTCTGTUbiquitin: 5′ to 3′Forward: TTCCGACACCATCGACAATGTReverse: CGACCATCCTCAAGCTGCTT

Total RNA was extracted from potato leaves using the RNeasy Plant Mini-Kit (Qiagen, Hilden, Germany) according to the manufacturer’s instructions. Total RNA was extracted from potato tubers as described [23]. After extraction, RNA was measured using a NanoDrop (Thermo Fisher Scientific, Waltham, MA, USA), and, subsequently, 1000 ng of RNA was taken to be treated with dsDNase before undergoing reverse transcription with a Maxima cDNA Kit (Thermo Fisher Scientific, Waltham, MA, USA) to generate a first-strand cDNA template. Ubiquitin was used as a reference gene while analyzing isoamylase gene expression. Additionally, triplicate NTC samples for each gene were used as negative controls. cDNA, diluted tenfold to a concentration of 5 ng/uL for each biological replicate, was amplified in triplicate using the above primers and with 2x qPCRBio SyGreen Blue Mix Hi-ROX Master Mix (PCRBio, London, England, UK) on a qTower3 system and qPCRSoft 3.4 software (Analytik Jena, Jena, Germany). The thermal profile was as follows: 95 °C for 2 min, and 45 cycles of 95 °C for 10 s and 60 °C for 30 s, followed by a melting curve analysis of 60 °C to 95 °C for 15 s with increments of 1 °C.

### 4.5. Starch Granule Size Analysis

Starch granules were analyzed in a similar manner as previously described [7]. After blending, 5 g of blended potato was combined with 5 mL of deionized water, vortexed until starch granules visibly started falling to the bottom of the tube, and centrifuged at 10,000 rcf for 5 min. The layer of potato tissue was removed, the starch pellet was washed, and the process was repeated twice. Iodine solution (0.33% iodine, 0.66% potassium iodide, *w*/*v*) was then added to the separated and cleaned starch granules. This solution was vortexed until all granules were visibly stained.

Granules were diluted and pipetted onto microscope slides and analyzed with light microscopy; 7 micrographs of each biological replicate were taken using an Olympus U-CMAD3 microscope (Olympus, Tokyo, Japan), an Invenio 5DII microscope camera, and DeltaPix InSight software (DeltaPix, Hovedstaden, Denmark) at 4× optical zoom. The surface areas of individual granules were calculated with ImageJ [18] as previously described [7]. Approximately 1000 starch granules were analyzed per biological replicate and pooled for analysis on a per-plant basis.

### 4.6. Starch Content Determination

In a 50 mL tube (Corning, NY, USA), 25 g of de-ionized water was added to 0.25 g of freshly blended potato in triplicate for each biological replicate. As described [17], the mixture was vortexed and microwaved (De’Longhi, Treviso, Italy) at 100% power (1100 Watts) for 200 s. After cooling, any of the deionized water that evaporated was added back until the tube was at its original weight. Extracts were sonicated with a probe sonicator for 5 min with 1 s-on 1 s-off pulses at 60% of the amplitude. The solution was then passed through a 0.45 µm filter, and diluted 1×, 5× and 25× into 1 mL aliquots in disposable cuvettes, to which 0.2 mL of iodine solution was added. The solution was mixed well and the absorbance was measured with a spectrometer at 600 nm to measure total starch.

A calibration curve was created with pure potato starch from Merck (Rahway, NJ, USA) in concentrations ranging from 0.01 mg/mL to 0.15 mg/mL as well as a blank. Absorbance was measured and a linear fit was derived and used to calculate the concentration.

### 4.7. Protein Extraction and Determination

Protein was extracted from the ground-up tubers using 0.0625M Tris-HCl, pH 6.8, and 2% SDS as previously described [3], in triplicate, for each biological replicate. Moreover, 1 g of ground potato was combined with 6 mL of extraction buffer and samples were spun at 4 °C for 4 h. Afterward, samples were centrifuged at 10,000 rcf for 10 min and the supernatant was collected for protein determination. Samples were kept at −20 °C until analysis. Protein electroporation in SDS-polyacrylamide gel (SDS-PAGE) was performed as previously described [24].

Total protein was quantified by the bicinchoninic assay (BCA) [25] using a Cyanagen µQPRO micro BCA kit (Bologna, Italy). The assay was performed on a 96-well plate (SPL Life Sciences, Gyeonggi-do, Republic of Korea) and performed on a spectrometer (Tecan, Maennedorf, Switzerland) at 562 nm.

A calibration curve was created by using BSA at concentrations of 20–2000 µg/mL and a blank. Absorbance was measured; a linear fit was derived and used to calculate the concentration.

### 4.8. Water Content Determination

A 50 mL tube was weighed, filled halfway with ground potato tubers, and then weighed again. The cap was replaced with a Kim-Wipe (Kimberly Clark, TX, USA) and a rubber band. The tube was frozen at −80 °C and then lyophilized for 48 h. The tubes were weighed again after lyophilization and dry weight was recorded. Water content was calculated by subtracting the dry weight from the fresh weight, and dividing the result by the fresh weight.

### 4.9. Sugar Extraction and Determination

A total of 30 milligrams of lyophilized tubers were weighed out in triplicate from each biological replicate and extracted using a mortar and pestle with 1.5 mL of 80% ethanol in order to extract soluble sugars, centrifuged at 1600 rcf for 10 min, and incubated at 70 °C for 30 min to get rid of invertase, as previously described [15,26].

Sulfuric acid was slowly diluted on ice to 72%. Moreover, 200 milligrams of anthrone were dissolved in 100 mL of the diluted sulfuric acid solution, on ice, and with stirring. In glass tubes, the sample was diluted with deionized water 10-fold to 1 mL and 4 mL of the anthrone solution, and added on ice. The samples were vortexed and chilled for 30 min prior to heating. The samples were heated for 15 min in a hot water bath at 80 °C.

After 15 min, the samples were cooled to room temperature and measured for absorbance at 620 nm. A calibration curve was created by using sucrose at concentrations of 5–25 µg/mL and also a blank. Absorbance was measured and a linear fit was derived and used to calculate the concentration.

### 4.10. Observation of the Sprouting Behavior

Potatoes set aside for the sprouting observation were kept separated by biological replicates in brown paper bags in the dark at room temperature for the entire duration of the experiment. Between 2 and 4 similarly sized tubers were used for each biological replicate (n = 20–30). Potatoes were monitored for over 120 days until all tubers reached 100% sprouting. A potato was considered to have sprouted when sprouts were over 2 mm in length as described [7]. The number of sprouts was calculated once 100% germination was achieved in the control and treated groups.

## 5. Conclusions

In conclusion, this experiment attempted to suppress the three isoamylase isoforms using repeated applications of a synthesized dsRNA-lmPEI foliar spray. It was hypothesized that reduced levels of starch, reduced starch granule sizes, increased protein, and an increase in sucrose with an earlier sprouting phenotype would be observed, as was observed in a different experiment with transgenic hairpin RNAi [7]. However, silencing of the genes was less efficient (Figure 1 and Figure 2), and the level of starch was not decreased (Figure 3). Despite this, a transgenic phenotype was achieved, which consisted of increased sucrose levels (Figure 4), smaller median starch granule sizes (Figure 5), and earlier sprouting with more sprouts per tuber (Figure 8 and Figure 9).

The phenotype achieved in this experiment is important as it shows the initial feasibility in actively changing agricultural crop metabolism via SIGS and without transgenic modification. When considering the ease of adoption and scaling up for field agricultural uses, RNA payload protection is critical [10]. The novel delivery system consisting of lmPEI was crucial in protecting RNA against damage from ribonucleases, temperature swings, and other environmental factors, as well as ensuring stability and slow release over time [9].

Additionally, the suppression of ISA3 in the tubers (without the suppression of ISA1 and ISA2) is significant for two reasons. First, it hints at the possibility that ISA3 may also be responsible for starch granule sizing in addition to its known roles of starch degradation and mobilization [5,6]. Second, it leads to further investigation into the capabilities and limitations of SIGS and the various administration methods of dsRNA.

SIGS has the ability to rapidly transform agriculture, with a much shorter time to the market and better acceptance than genetic transformation techniques [10]. Targets can be rapidly designed and adapted based on environmental factors or the specific needs of a crop. Further research is needed to examine bio-synthetic pathways that can be altered via exogenous dsRNA. Experiments such as this are important to show the feasibility and highlight potential barriers to SIGS applications. Areas needing more research include molecule mobilization within the plant, payload delivery, carrier components, and effective sequence design. Further experimentation could lead to the usage of the platform in targeting other commercially-relevant tuberous genes [27] and various use cases in increasing sucrose across a number of different crops that contain the isoamylase gene family, such as rice [28], barley [29], cassava [30], and sweet potatoes [31].

## Figures and Tables

**Figure 1 ijms-24-00190-f001:**
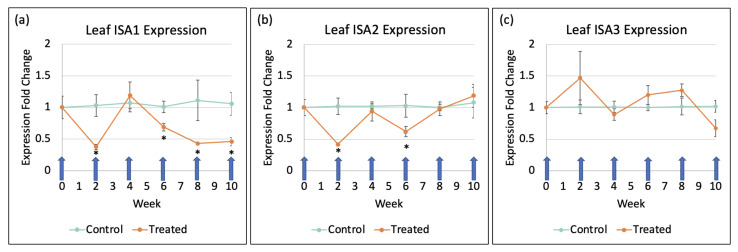
ISA gene expression in leaves over the course of the growth cycle. Application of foliar dsRNA-lmPEI is denoted by the blue arrows. Samples for PCR were collected prior to the application of the particles. Significantly different (*p* ≤ 0.05) expression levels between treatment groups are denoted with ‘*’ (n = 15 plants, in triplicate, each for treatment and control). (**a**) ISA1, (**b**) ISA2, (**c**) ISA3.

**Figure 2 ijms-24-00190-f002:**
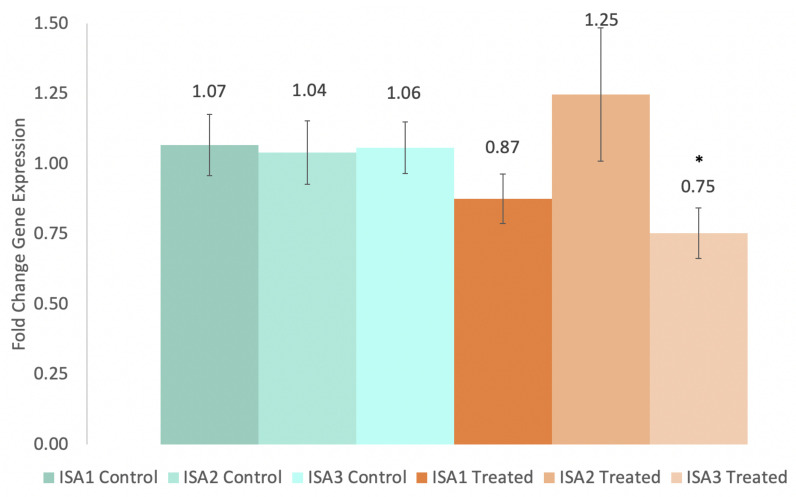
ISA gene expression in tubers at harvest. Significantly different (*p* ≤ 0.05) expression levels between treatment groups are denoted with ‘*’. (n = 15 plants, in triplicate, each for the treatment and control).

**Figure 4 ijms-24-00190-f004:**
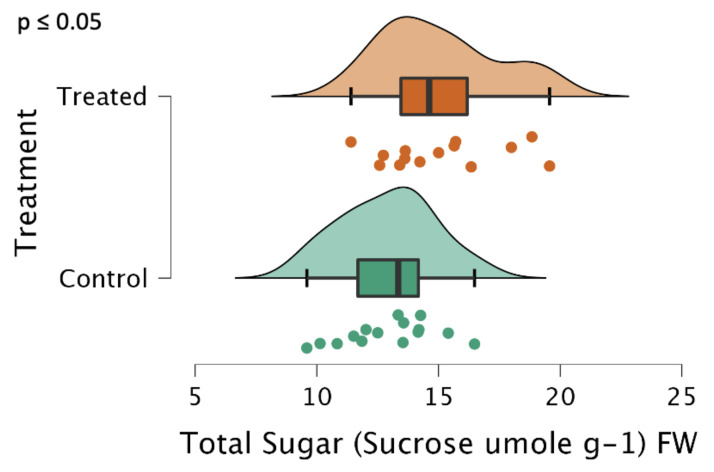
Total sugars, reported in sucrose equivalents, represented as a raincloud plot [16]. The graph is comprised of a boxplot with the lines representing the median, lower, and upper-end measurements, as well as the lower and upper quartiles. Each dot represents the recorded value of the biological replicate, with the cloud representing the overall distribution of the data. Note that the differences were significant, *p* ≤ 0.05 (n = 15 plants, in triplicate, each for the treatment and control). Sugars were extracted from blended and lyophilized tuber tissues using ethanol and then determined spectrometrically with sulfuric acid and anthrone, using a calibration curve with sucrose.

**Figure 5 ijms-24-00190-f005:**
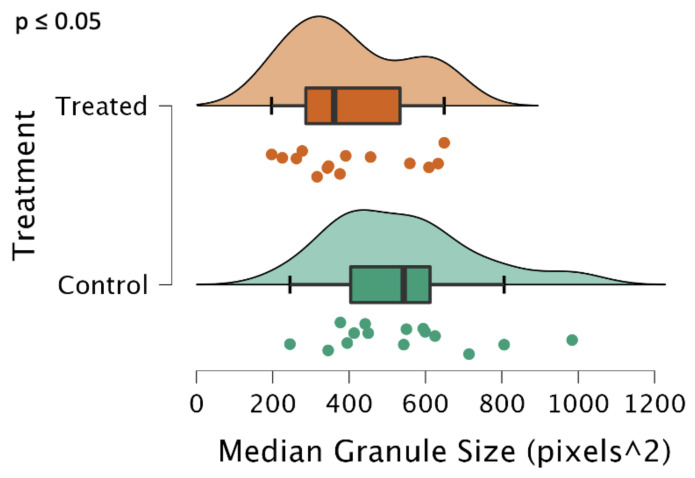
Median granule size, reported in the area of pixels as determined by ImageJ [18], represented as a raincloud plot [16]. The graph is comprised of a boxplot with lines representing the median, lower, and upper-end measurements, as well as lower and upper quartiles. Each dot represents the recorded value of the biological replicate, with the cloud representing the overall distribution of the data. Each dot on the raincloud plot represents a biological replicate (n = 15 plants each for treatment and control), for which approximately 1000 separate starch granules were analyzed for each plant. Note the differences were significant, *p* ≤ 0.05.

**Figure 6 ijms-24-00190-f006:**
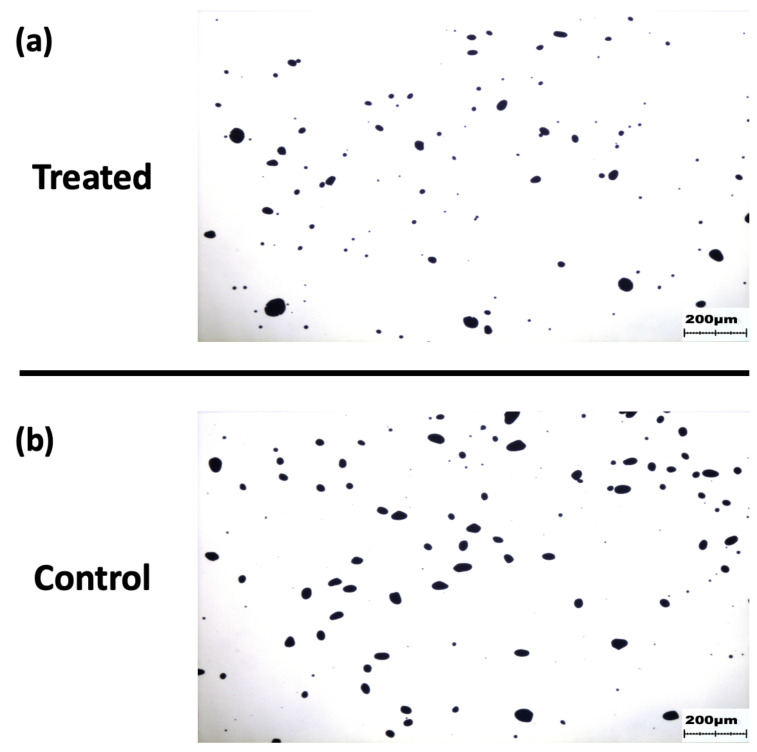
Representative micrographs of starch granules stained with iodine for (**a**) treated and (**b**) control groups, captured at 4× optical zoom.

**Figure 7 ijms-24-00190-f007:**
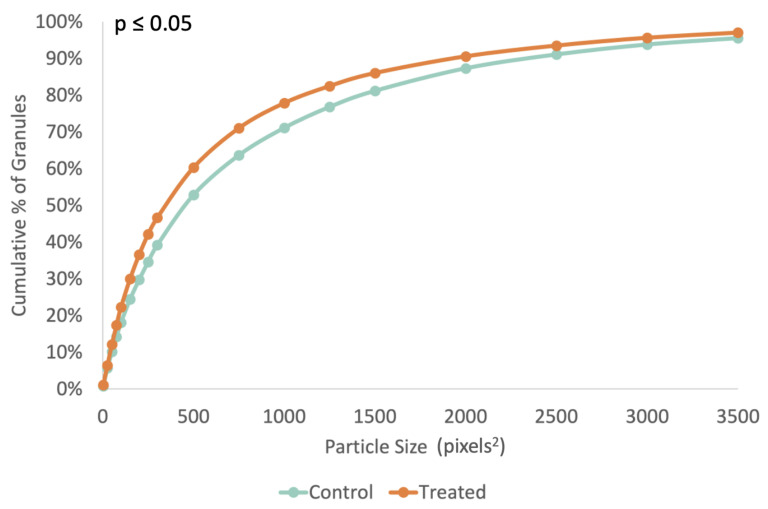
Cumulative distribution of starch granules according to the surface area, in pixels, as determined by ImageJ [18]; (n = 15,000 for control and n = 14,000 for treated). Note the left-shift toward a larger proportion of smaller granules in the treated group.

**Figure 8 ijms-24-00190-f008:**
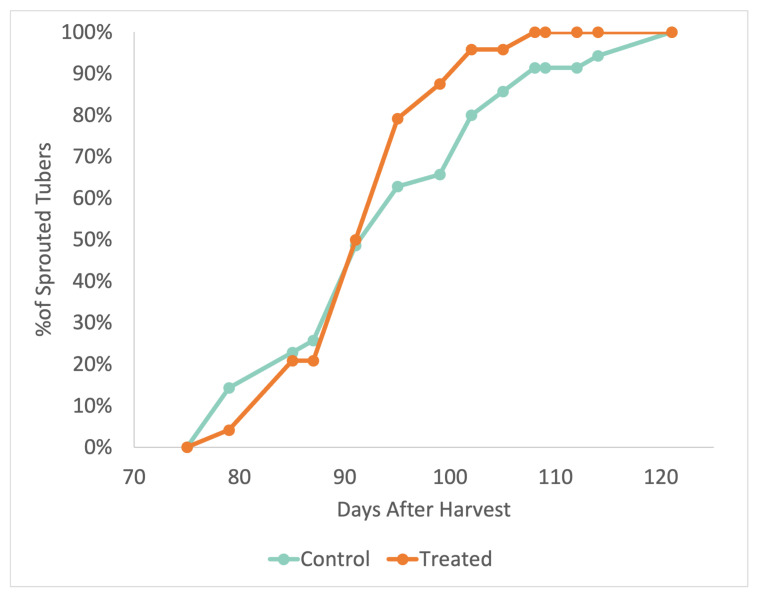
A comparison of the treated and control groups sprouting behavior over the course of 120 days (n = 1–4 potatoes per biological replicate, 34 potatoes examined for control and 24 for treated). A biological replicate was considered to be at 100% sprouting when its sprouts reached a length of over 2 mm as previously described [7].

**Figure 9 ijms-24-00190-f009:**
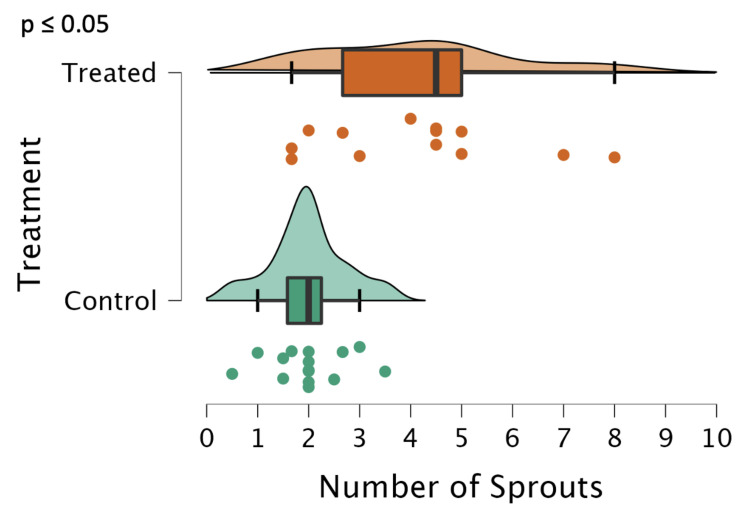
The number of sprouts emerging from each tuber, represented as a raincloud plot [16]. The graph is comprised of a boxplot with the lines representing the median, lower, and upper-end measurements, as well as lower and upper quartiles. Each dot represents the recorded value of the biological replicate, with the cloud representing the overall distribution of the data; (n = 34 for control, 24 for treated). The sprouts were counted once a group reached 100% germination across all biological replicates [7].

## Data Availability

Data are available upon request.

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
