# Peer review of "Foliar Application of dsRNA Targeting Endogenous Potato (Solanum tuberosum) Isoamylase Genes ISA1, ISA2, and ISA3 Confers Transgenic Phenotype"

_ijms, 2022, doi:10.3390/ijms24010190_

Round 1
Reviewer 1 Report
Reviewer’s comments
The manuscript entitles Foliar Application of dsRNA Targeting Endogenous Potato (Solanum tuberosum) Isoamylase Genes ISA1, ISA2, and ISA3 Confers Transgenic Phenotype” described the application of dsRNA targeting 3 genes involve in starch metabolism in potato. The work has been carried out with several techniques and the manuscript is well-structured.
However, there are several major points which I would like the authors to
1. Please provide the experimental outline as figure describing the detail for each experiment. In the current form, I could not figure out how many plants/tubers/leaves were used in each biological replicate.
2. With the spray, how the authors control consistency between each spray.
3. Does the absorption of the dsRNA is determined?
4. Do the authors analyze the characteristic of the nanoparticle? For instance, size, PDI and zeta-potential? All of these factors are important for the nanoparticle.
5. For statistical analysis, please analyze based on p-value ≤ 0.05 throughout the whole manuscript.
6. Please provide the figure of the construct used in this work.
7. Has the specificity of the dsRNA been determined? Do the authors have scrmble construct?
8. Please remove the description of protein analysis in 2.7. It is too lengthy. Please provide only the reference.
9. When describe the concentration as range, please put the lower concentration first i.e. 20-2000 microgram/mL
10. For experimental design, the dsRNAs were applied every two week and the samples were collected prior to the application. Why the authors collect the samples before the spray? Did the authors test stability of the dsRNA, the absorption and how long does the action of the dsRNA last?
11. Figure 1, please display the figure with p-value ≤ 0.05.
12. Figure 2, the difference between ISA3 control and ISA3 treatment is rather small. Please provide the absolute value of this experiment.
13. Figure 2, when n=15. Please clarify n number is form15 plants, 15 tubers?
14. Figure 6 and 7 should be combined, labelled, and choose only representative figure to display. In addition, the scale bar should be included here.
15. Please state the p-value of figure 8
Reviewer 2 Report
In the manuscript " Foliar Application of dsRNA Targeting Endogenous Potato (Solanum tuberosum) Isoamylase Genes ISA1, ISA2, and ISA3 Confers Transgenic Phenotype’ Simon and collaborators evaluated the efficiency of a novel delivery system with foliar applications of dsRNA to achieve a transgenic phenotype in potato.
Overall, the manuscript is very well written, clear, and well organized. The Materials and Methods section describes in detail the methodologies used which are also appropriate to accomplish the objectives of the study. The Results section presents clearly the results being well interpreted. The Discussion section makes a good discussion of the results obtained. The conclusions are very interesting and well presented.
Additional comments:
The authors should include the primers efficiency and amplicon sizes.
Line 177 and throughout the text: replace qRT-PCR to RT-qPCR
Line 179: place ncbi accession number for ubiquitin gene
Line 2014: change to: The thermal profile was as follows: 95°C for 2 minutes, and 45 cycles of 95°C for 10 seconds and 60°C for 30 seconds, followed by a melting curve analysis of 60°C to 95°C for 15 seconds with increments of 1°C.
Round 2
Reviewer 1 Report
The reviewer would like to thank authors for theirs responses. The manuscript has been revised point-by-point as suggested.